# Could prokinetic agents protect long-term nasogastric tube-dependent patients from being hospitalized for pneumonia? A nationwide population-based case-crossover study

Kun-Siang Huang[1], Bo-Lin Pan[1], Wei-An Lai[1], Pin-Jie Bin[1], Yao-Hsu Yang[2,3,4], Chia-Pei Chou[1] *

1 Department of Family Medicine, Kaohsiung Chang Gung Memorial Hospital and Chang Gung University College of Medicine, Kaohsiung, Taiwan, 2 Department of Traditional Chinese Medicine, Chang Gung Memorial Hospital, Chiayi, Taiwan, 3 Health Information and Epidemiology Laboratory of Chang Gung Memorial Hospital, Chiayi, Taiwan, 4 School of Traditional Chinese Medicine, College of Medicine, Chang Gung University, Taoyuan, Taiwan

* libra760924@cgmh.org.tw

**Data Availability Statement:** All relevant data are within the manuscript and its Supporting Information files.

## Abstract

### Background

Some studies have indicated that the use of prokinetic agents may reduce pneumonia risk in some populations. Nasogastric tube insertion is known to increase the risk of pneumonia because it disrupts lower esophageal sphincter function. The aim of this study was to evaluate whether prokinetic agents could protect long-term nasogastric tube-dependent patients in Taiwan from being hospitalized for pneumonia.

### Methods

A case-crossover study design was applied in this study. Long-term nasogastric tube-dependent patients who had a first-time admission to a hospital due to pneumonia from 1996 to 2013 that was recorded in the Taiwan National Health Insurance Research Database were included. The case period was set to be 30 days before admission, and two control periods were selected for analysis. Prokinetic agent use during those three periods was then assessed for the included patients. Conditional logistic regression was used to calculate the odds ratio (OR) for pneumonia admission with the use of prokinetic agents.

### Results

A total of 639 first-time hospitalizations for pneumonia among patients with long-term nasogastric tube dependence were included. After adjusting the confounding factors for pneumonia, no negative association between prokinetic agent use and pneumonia hospitalization was found, and the adjusted OR was 1.342 (95% CI 0.967–1.86). In subgroup analysis, the adjusted ORs were 1.401 (0.982–1.997), 1.256 (0.87–1.814), 0.937 (0.607–1.447) and 2.222 (1.196–4.129) for elderly, stroke, diabetic and parkinsonism patients, respectively.

**Funding:** This work was supported by grants from Chang Gung Memorial Hospital (CFRPG8H0321).

**Competing interests:** The authors have declared that no competing interests exist.

## Conclusion

Prokinetic agent use had no negative association with pneumonia admission among long-term nasogastric tube-dependent patients in Taiwan.

## Introduction

Pneumonia was the third leading cause of death in Taiwan in 2018, with the number of deaths from pneumonia increasing by 7.5% over the number in 2017 [1]. There are several known risk factors for pneumonia, including unclear consciousness, dementia, Parkinson's disease, stroke, chronic obstructive pulmonary disease, gastroesophageal reflux disease, gastroparesis, bowel obstruction and ileus, esophageal motility disorders, the presence of an endotracheal tube and enteral tube feeding [2–7]. Enteral feeding with nasogastric tube use is indicated for patients with impaired swallowing function and malnutrition. In Taiwan, long-term nasogastric tube feeding may be required in patients with stroke, dementia, parkinsonism and old age due to the progression of neurodegenerative diseases or aging-related functional declines. However, enteral feeding with nasogastric tube placement may disrupt the function of the upper and lower esophageal sphincters, increase the frequency of transient lower esophageal sphincter relaxations and desensitize the pharyngoglottal adduction reflex [3]. Relatedly, aspiration pneumonia is one potential consequence of the inhalation of oropharyngeal secretions or gastric contents [8], and nasogastric tube placement may increase the risk of both aspiration and pneumonia [9] due to the aforementioned mechanism. Meanwhile, prokinetic agents such as metoclopramide, cisapride, mosapride and domperidone, which are used for gastroesophageal reflux and gastroparesis, increase gastric motility and facilitate gastric emptying [10–13], and in a prior study, it was found that metoclopramide may decrease the pneumonia risk of acute stroke patients with nasogastric tube feeding [14]. Until now, however, there had been no studies investigating the association between the use of prokinetics and pneumonia in long-term nasogastric tube-fed patients. Therefore, the objective of this study was to examine whether prokinetic use protects long-term nasogastric tube-fed patients from being hospitalized for pneumonia.

## Materials and methods

### Data source

Begun in 1995, Taiwan's National Health Insurance (NHI) system currently provides healthcare services and coverage to more than 99% of Taiwan's residents. The beneficiary data of the people covered by the NHI system is recorded in the National Health Insurance Research Database (NHIRD), which is maintained by the National Health Research Institutes. The present study was conducted using data from the Longitudinal Health Insurance Database 2005 (LHID 2005), a subset of the NHIRD that includes the longitudinal health care data for NHI beneficiaries for the period from 1996 to 2013. More specifically, the LHID 2005 contains data for 1 million representative beneficiaries sampled in 2005 from among the overall total of roughly 25.68 million people registered with the NHI system, with the data in question including the given patient's sex; age; diagnoses according to International Classification of Diseases, Ninth Revision, Clinical Modification (ICD-9-CM) codes; prescribed medications; medication dosages; durations of prescribed medications; medical interventions and medical expenditures.

## Patient selection

The study covered the period from January 1, 1996, to December 31, 2013. The discharge diagnoses of interest in the LHID 2005 consisted of 5 ICD-9-CM diagnoses. In the first step of the patient selection process, patients aged over 20 years old were included if they met either of the following criteria: 1. a primary discharge diagnosis of aspiration pneumonia (ICD-9-CM 507) or pneumonia (ICD-9-CM 480–487) or 2. a primary discharge diagnosis of septicemia (ICD-9-CM 038.0–039.9), respiratory failure (ICD-9-CM 518.81–518.89) or sepsis (ICD-9-CM 995.91–995.92) in combination with a secondary discharge diagnosis of aspiration pneumonia (ICD-9-CM 507) or pneumonia (ICD-9-CM 480–487). In a prior study, the diagnostic codes for pneumonia identification were validated [15], and only one modification was applied in this study (specifically, ICD-9-CM 507 was added as a secondary discharge diagnosis). If a patient had several admissions with discharge diagnoses that met the above criteria, only the first-time admission for pneumonia was included to avoid interference during the control periods. The first-time admission date of each patient was defined as the index date. In Taiwan, a nasogastric tube is usually replaced at least once every month depending on the condition of the nasogastric tube and the patients' long-term nasogastric tube dependence was presumed to be at least 3 months in this study. Therefore, in the second step of the patient selection process, the patients with long-term nasogastric tube use were selected by screening for the procedure code for nasogastric tube insertion to see if it was noted in the LHID 2005 data at least 1 time per month in the 7 months before the index date to ensure that each included patient was nasogastric tube dependent at least 3 months among the case period and the control periods. Meanwhile, any patients who received a gastrostomy or jejunostomy before the index date were excluded. The study patient selection flow chart is shown in Fig 1.

## Study design

The association between the use of prokinetics and hospitalization for pneumonia was assessed using a case-crossover study design. Maclure et. al proposed the case-crossover design as a means of providing within-subject comparisons of transient effects for acute events [16]. Because the case-crossover design uses each patient as his or her own control group by considering data from a different time point or time points, the confounding factors are automatically adjusted for [17, 18]. In this study, the case period was defined as a time period before the index date, and the control periods were defined as time periods without any pneumonia admission. Each patient's exposure to prokinetics during the case period was compared to the patient's exposure to prokinetics during the control periods. In Taiwan, prokinetics are usually indicated for nausea, vomiting, gastroesophageal reflux and poor digestion, and physicians may prescribe these medicines for a duration of 28 to 30 days in stable patients. As such, the time periods for the case period, control periods and washout period were defined as 30 days in this study. Two control periods (specifically, the periods 91 to 120 days and 121 to 150 days before admission) were selected to compare with a single case period. The study design is shown in Fig 2. In this study, sensitivity analyses were conducted by changing the length of each time period to 14 days and 45 days because patients may not always use a given medicine regularly. The length of the washout period was also changed to 7 days and 15 days to assess the robustness of this study in terms of sensitivity analyses.

## Variable assessment and confounding factors

The prescription status for prokinetics during the case period and control periods was examined for each patient. The prokinetics themselves were identified by the Anatomical Therapeutic Chemical (ATC) code A03F (propulsives). The data indicate each patient's sex, age,

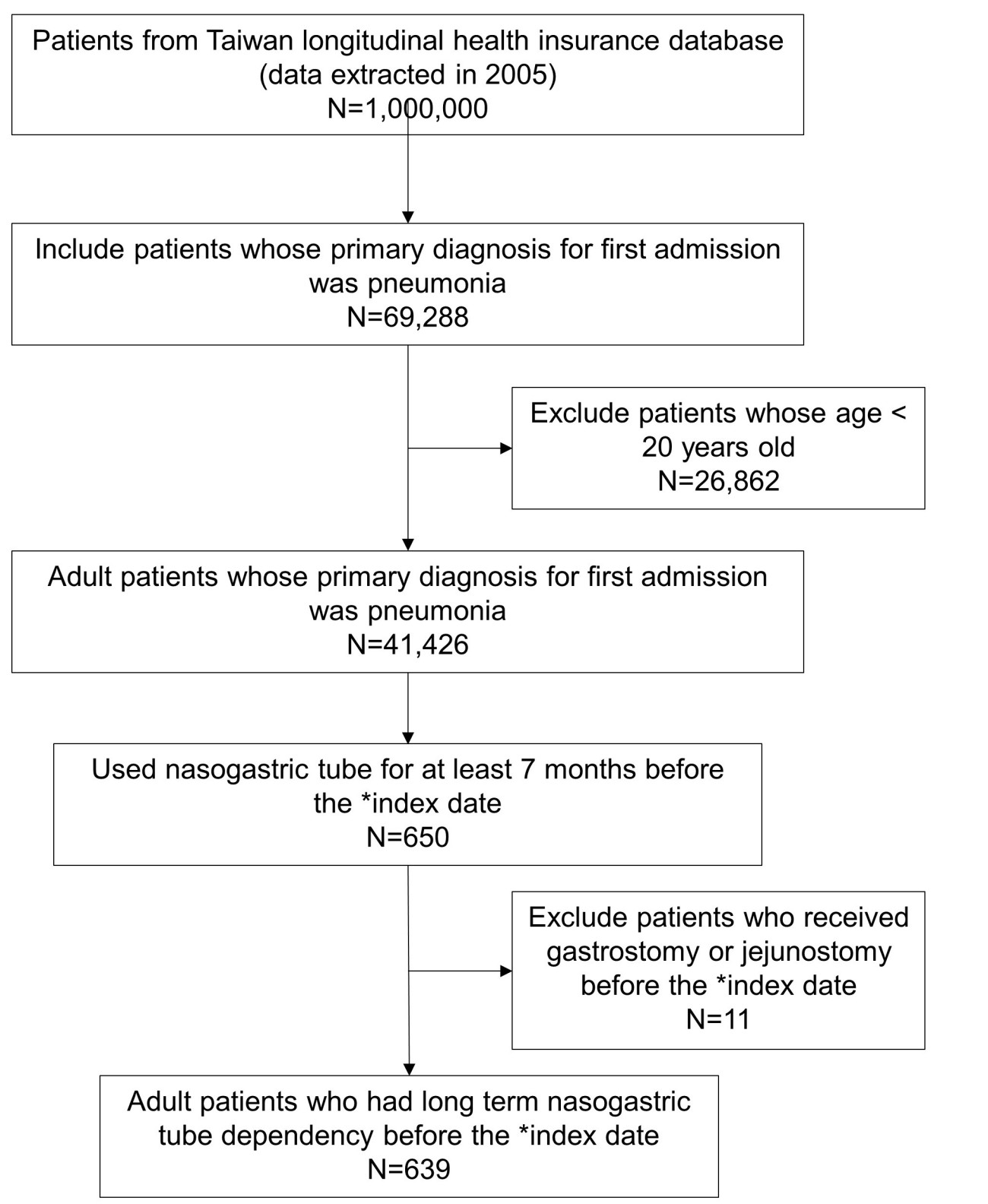

**Fig 1. The flow diagram of the patient selection process.** *Index date refers to the first date of each patient's first admission with the primary diagnosis of pneumonia in the LHID 2005.

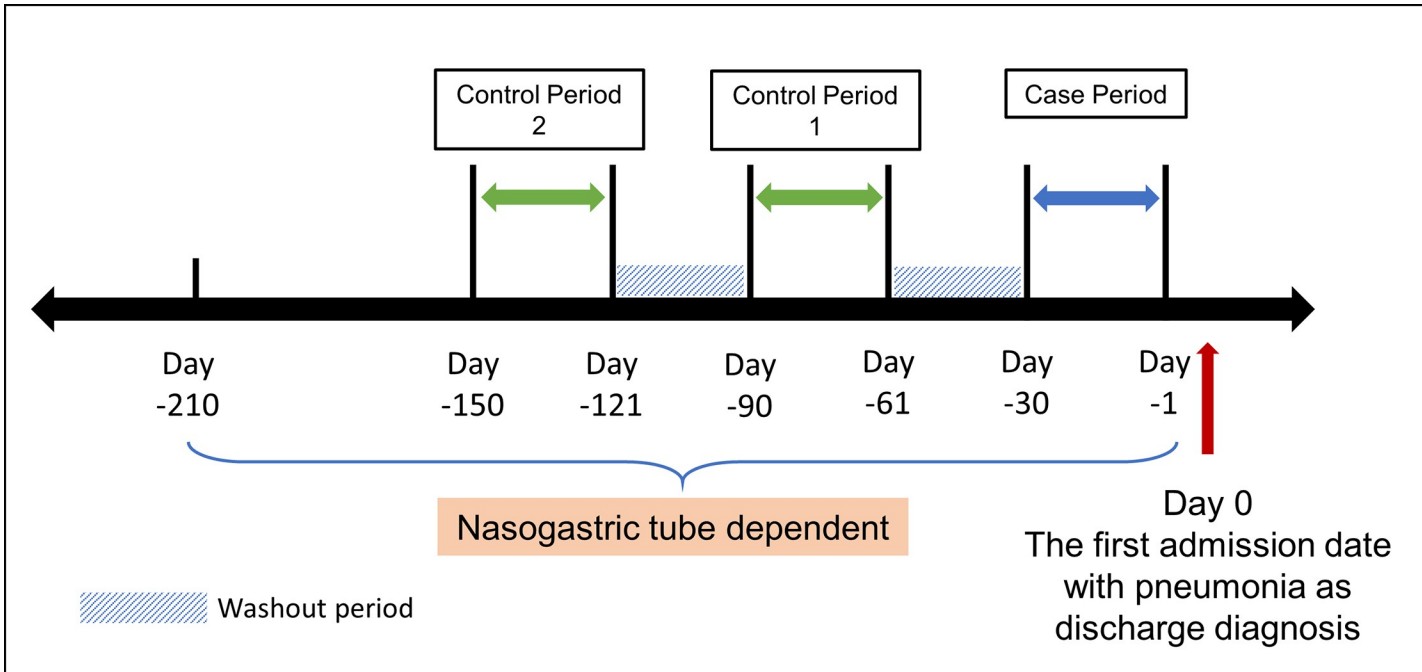

**Fig 2. The 2:1 case-crossover study design used in the primary analysis.** Day 0 indicates the first admission date with pneumonia as the discharge diagnosis. The time period and washout periods each lasted 30 days.

comorbidities and medications prescribed were also collected. Aside from clinical characteristics, some medications have been reported to be associated with pneumonia occurrence. To account for the possible impacts of these medications, they were regarded as confounding factors and their prescription statuses were also examined during the case period and control periods. The confounding medications that were included are as follows: 1. Antipsychotic agents (ATC code: N05A): In prior studies, antipsychotic agents were reported to be associated with increased pneumonia risk due to unknown mechanisms [19–21]; 2. Benzodiazepine-receptor agonists (BZRA, ATC codes: A05CD, A05BA, N03AE and N03CF): BZRAs were also reported to increase the risk of aspiration pneumonia by reducing consciousness [22–24]; 3. Histamine H2-blockers (H2B, ATC code: A02BA) and proton pump inhibitors (PPI, ATC code: A02BC): H2Bs and PPIs suppress gastric acid secretion, which may facilitate pathogen colonization in the upper gastrointestinal tract and subsequently cause pneumonia by aspiration. In fact, the aforementioned acid-reducing agents have been proved to increase the risk of pneumonia in several studies [25–29]; 4. Statin (ACT codes: C10AA, C10BA, C10BX): Statins have been reported to reduce the risk of pneumonia due to possible anti-inflammatory and immunomodulatory effects [30–32]; 5. Angiotensin receptor blockers (ARB, ATC codes: C09C, C09D) and angiotensin-converting enzyme inhibitors (ACEi, ATC codes: C09A, C09B): ARBs and ACEis have also been found to be associated with reduced pneumonia risk [33]. The abovementioned medication exposures were regarded as confounding factors in the adjusted analyses.

### Statistical analysis

The SAS software version 9.4 was used for all the statistical analyses. Conditional logistic regression was applied for paired data (1 case period matched to 2 control periods), and odds ratios (ORs) and 95% confidence intervals (CIs) for pneumonia hospitalization associated with

exposure versus non-exposure to prokinetics were calculated. In addition to crude ORs, ORs adjusted for exposure to antipsychotic agents, BZRAs, H2Bs, PPIs, statins, ARBs and ACEis were also calculated. Subgroup analyses were performed for patients with diabetes, stroke, parkinsonism and old age (age≧65 years old). A p value less than 0.05 was considered to be statistically significant.

### Ethics statement

The study was implemented with the approval of the Chang Gung Medical Foundation Institutional Review Board (201801143B0). Proof of informed consent documents was not required by the review board because the NHIRD had de-identified all patient data.

### Results

The basic characteristics of the included patients are shown in Table 1. A total of 639 long-term nasogastric tube-dependent patients with pneumonia admission were included in the analysis. The included patients' mean age was 77.91±12.06 years, 88% were older than 65 years old and 54% were women. Furthermore, 80% had a history of stroke, 61% had a history of diabetes mellitus and 26% had a history of parkinsonism. In total, 124 patients used metoclopramide, 56 used mosapride and 88 used domperidone during the case or control periods.

Table 2 lists all the analysis results of this study. In the primary analysis, the crude OR (95% CI) of the study population for all prokinetics exposure and pneumonia admission was 1.293 (0.941–1.777). After adjustment for confounding factors, the adjusted OR (95% CI) was 1.342 (0.967–1.86). In the further analysis of individual prokinetic drugs for the study population, the adjusted ORs (95% CI) were 1.03 (0.672–1.578), 1.53 (0.814–2.877) and 1.394 (0.838–2.32) for metoclopramide, mosapride and domperidone use, respectively. In the subgroup analysis, the prokinetics exposure was not significantly associated with reduced hospitalization for pneumonia among the elderly, diabetes mellitus, stroke or parkinsonism patients. The adjusted ORs (95% CI) were 1.673 (1.052–1.661) and 2.222 (1.196–4.129) in females and parkinsonism patients, respectively, which indicated a slight positive association between prokinetics exposure and hospitalization for pneumonia. In the sensitivity analyses (S1–S4 Tables) for different time periods and washout periods, similarly, there was no significant reduction in

**Table 1. Patient demographic and clinical characteristics, n = 639.**

| Variables | Number (%) |
|---|:---:|
| Age | |
| Age ≥ 65 years old | 564 (88) |
| Age < 65 years old | 75 (12) |
| Gender | |
| Female | 348 (54) |
| Male | 291 (46) |
| Comorbidity | |
| Diabetes mellitus | 391 (61) |
| Stroke | 510 (80) |
| Parkinsonism | 166 (26) |
| Prokinetic agents | |
| Metoclopramide user | 124 (19) |
| Mosapride user | 56 (9) |
| Domperidone user | 88 (14) |

**Table 2. The association between prokinetics exposure and pneumonia admission.**

| | | Crude OR | 95% Cl | | P value | Adjusted OR[a] | 95% Cl | | P value |
|---|---|---|---|---|---|---|---|---|---|
| General population | All prokinetics | 1.29 | (0.94 | 1.78) | 0.1133 | 1.34 | (0.97 | 1.86) | 0.0782 |
| n = 639 | Metoclopramide | 1.03 | (0.68 | 1.56) | 0.8868 | 1.03 | (0.67 | 1.58) | 0.8924 |
| | Mosapride | 1.48 | (0.80 | 2.72) | 0.2081 | 1.53 | (0.81 | 2.88) | 0.1865 |
| | Domperidone | 1.30 | (0.79 | 2.14) | 0.3010 | 1.39 | (0.84 | 2.32) | 0.2013 |
| Age ≧ 65 years old | All prokinetics | 1.35 | (0.96 | 1.90) | 0.0872 | 1.40 | (0.98 | 2.00) | 0.0627 |
| n = 564 | Metoclopramide | 1.07 | (0.69 | 1.68) | 0.7582 | 1.06 | (0.67 | 1.68) | 0.8122 |
| | Mosapride | 1.43 | (0.76 | 2.72) | 0.2714 | 1.47 | (0.76 | 2.86) | 0.2556 |
| | Domperidone | 1.42 | (0.83 | 2.43) | 0.1955 | 1.54 | (0.89 | 2.67) | 0.1225 |
| Male | All prokinetics | 1.02 | (0.64 | 1.62) | 0.9369 | 1.04 | (0.65 | 1.67) | 0.8643 |
| n = 291 | Metoclopramide | 0.88 | (0.47 | 1.63) | 0.6768 | 0.85 | (0.45 | 1.61) | 0.6268 |
| | Mosapride | 1.15 | (0.48 | 2.76) | 0.7632 | 1.19 | (0.47 | 3.00) | 0.7145 |
| | Domperidone | 0.78 | (0.36 | 1.68) | 0.5262 | 0.84 | (0.38 | 1.83) | 0.6581 |
| Female | All prokinetics | 1.61 | (1.03 | 2.50) | 0.0350* | 1.67 | (1.05 | 2.66) | 0.0296* |
| n = 348 | Metoclopramide | 1.18 | (0.67 | 2.08) | 0.5604 | 1.17 | (0.65 | 2.12) | 0.5968 |
| | Mosapride | 1.89 | (0.80 | 4.45) | 0.1456 | 1.98 | (0.82 | 4.78) | 0.1275 |
| | Domperidone | 1.98 | (1.00 | 3.89) | 0.0487* | 2.11 | (1.05 | 4.24) | 0.0352* |
| Diabetes Mellitus | All prokinetics | 0.91 | (0.60 | 1.39) | 0.6696 | 0.94 | (0.61 | 1.45) | 0.7707 |
| n = 391 | Metoclopramide | 0.71 | (0.41 | 1.24) | 0.2330 | 0.73 | (0.41 | 1.28) | 0.2677 |
| | Mosapride | 0.90 | (0.41 | 1.98) | 0.7905 | 0.82 | (0.36 | 1.87) | 0.6412 |
| | Domperidone | 0.93 | (0.50 | 1.75) | 0.8321 | 1.03 | (0.54 | 1.96) | 0.9347 |
| Stroke | All prokinetics | 1.19 | (0.84 | 1.70) | 0.3306 | 1.26 | (0.87 | 1.81) | 0.2233 |
| n = 510 | Metoclopramide | 1.04 | (0.66 | 1.64) | 0.8756 | 1.07 | (0.67 | 1.71) | 0.7878 |
| | Mosapride | 1.41 | (0.73 | 2.72) | 0.3099 | 1.44 | (0.72 | 2.86) | 0.3009 |
| | Domperidone | 0.97 | (0.55 | 1.73) | 0.9221 | 1.06 | (0.59 | 1.92) | 0.8421 |
| Parkinsonism | All prokinetics | 2.41 | (1.32 | 4.40) | 0.0043* | 2.22 | (1.20 | 4.13) | 0.0115* |
| n = 166 | Metoclopramide | 1.91 | (0.87 | 4.18) | 0.1073 | 1.79 | (0.78 | 4.10) | 0.1695 |
| | Mosapride | 2.73 | (0.79 | 9.50) | 0.1138 | 2.80 | (0.66 | 11.94) | 0.1644 |
| | Domperidone | 2.00 | (0.82 | 4.89) | 0.1289 | 1.90 | (0.77 | 4.67) | 0.1639 |

CI = Confidence Interval, OR = Odds Ratio.

* p value < 0.05.

[a] Odds ratios adjusted for antipsychotic agents, benzodiazepine-receptor agonists, histamine H2-blockers, proton pump inhibitors, statins, angiotensin receptor blockers, and angiotensin-converting enzyme inhibitors exposure.

hospitalization for pneumonia with prokinetics exposure. Therefore, the analyses results are robust.

## Discussion

According to the analyses of this nationwide population-based study, there was no significant negative association between prokinetics exposure and hospitalization for pneumonia among the investigated long-term nasogastric tube-dependent population in Taiwan. In the subgroup analyses, there was also no significant negative association between hospitalization for pneumonia and prokinetics exposure among the elderly patients or patients with diabetes mellitus, stroke and parkinsonism.

In Taiwan, 198,393 people used nasogastric tubes in 2018, and of these people, 61.3% were older than 65 years old and 56.6% were males [34]. In this study, elderly patients accounted for 88% of the long-term nasogastric tube-dependent patients, and male patients accounted for

46% of the long-term nasogastric tube-dependent patients. The patients included in this study were nasogastric tube-dependent for at least 7 months before the occurrence of hospitalization due to pneumonia. Therefore, the patients included in this analysis, 80% of whom were stroke patients, 60% of whom were diabetes mellitus patients and 88% of whom were elderly patients, were at substantially high risk for pneumonia [35].

Hiyama et al. indicated that prokinetic agents may reduce the risk of aspiration pneumonia in tube-fed patients via the direct effects of their motility stimulation properties [36]. Warusevitane et al. found that among tube-fed acute stroke patients, a placebo group had a higher rate of pneumonia than a metoclopramide group (rate ratio 5.24, p<0.001), which indicated that metoclopramide use may prevent pneumonia from occurring. Pareek et al. indicated that among tube-fed patients with severe developmental disabilities, prokinetic therapy (cisapride) could reduce the rate of hospitalization for aspiration pneumonia, with the relative risk being reduced by 4.5 times [37]. However, in our analysis, there was no significant negative association between prokinetics use and hospitalization for pneumonia. Even among the stroke patients, the adjusted OR for all prokinetics was 1.265 (p = 0.2233), while that for metoclopramide was 1.067 (p = 0.7878). Because all the patients included in this case-crossover design study had relatively long-term nasogastric tube dependence, their general health conditions may have been quite different from those of post-acute stroke patients. Because the drug license for cisapride was withdrawn in Taiwan in 2004 due to the adverse effect of lethal QT prolonged syndrome, the number of cisapride users included in the primary analysis was small. Therefore, cisapride users were not included in this case-crossover study. A prokinetic agent with similar pharmacologic action as cisapride, mosapride, was more widely used among this Taiwanese cohort. However, there was still no significant negative association between mosapride use and pneumonia admission in this analysis.

Nassaji et al. indicated that metoclopramide appeared to have no effect on the occurrence of nosocomial pneumonia among patients with nasogastric feeding [38]. In another randomized control trial, Yavagal et al. also found that metoclopramide use did not decrease the rate of nosocomial pneumonia in critically ill patients undergoing nasogastric tube feeding [39]. Two meta-analysis studies also revealed no obvious protective effect of metoclopramide in critically ill tube-fed patients. In their meta-analysis, Lewis et al. indicated that the risk ratio of metoclopramide use compared with placebo for intensive care unit (ICU)-acquired pneumonia was 1.00 (95% CI: 0.76–1.32) [40]. Meanwhile, in a pooled meta-analysis, Liu et al. also reported that metoclopramide use showed no reduction in pneumonia risk compared with placebo in critically ill tube-fed patients, with the risk ratio being 0.79 (95% CI: 0.45–1.38) [41]. The abovementioned studies showed results similar to those of our study, that is, that prokinetics use does not reduce the occurrence of pneumonia. The patients included in our study were mostly elderly and were nasogastric tube-dependent for at least 7 months. The findings of our study thus provide further evidence that prokinetics agent use among elderly and chronic nasogastric tube-fed patients does not prevent pneumonia from occurring, despite the gastric-emptying properties of such agents.

In addition to the lack of a negative association between prokinetics exposure and pneumonia admission, this study found, in the subgroup analysis, that the adjusted ORs in female patients and patients with parkinsonism were 1.673 (95% CI: 1.052–2.661) and 2.222 (95% CI:1.196–4.129), respectively. There were also increased trends of positive association between prokinetics exposure and pneumonia admission in the overall population, elderly patients and stroke patients. Among possible reasons for these results, first, receptor hyperstimulation and automatic downregulation or upregulation mechanisms of synaptic receptors [42] may have played some role. Perhaps the downregulation of serotonin 5HT4 receptors after long-term mosapride exposure or the upregulation of dopamine receptors after long-term

metoclopramide or domperidone exposure resulted in the lack of a negative association between prokinetics exposure and pneumonia admission, although there is no relevant basic research in this field to date. Second, indication bias may also have played an important role in this data analysis. Indication bias occurs when results may be influenced by the reason for a prescription being related to the measured outcome [43, 44]. Even though the case-crossover design eliminated confounding factors, such as the comorbidities of the included patients, through within-subject comparison, indication bias may still have been present and could have interfered with the outcomes. In addition to fever, chills and productive cough, up to 20% of pneumonia patients have gastrointestinal symptoms such as nausea and vomiting, while some patients also have malaise and fatigue [45, 46]. What's more, patients with parkinsonism also have high rates of gastroparesis symptoms such as nausea, vomiting, early satiety and postprandial fullness [47–50]. Therefore, such patients may initially receive prescribed prokinetic agents for gastrointestinal symptoms in an outpatient department before being hospitalized due to pneumonia, a phenomenon which may have contributed an indication bias in this analysis.

## Limitations

Although this was the first nationwide database study to investigate the association between prokinetic agent exposure and pneumonia admission among long-term nasogastric tube-dependent patients, there were several limitations in the study design. First, the number of included patients was relatively small even though the large nationwide health insurance database analyzed in this study contained 1 million representative samples. Only slightly fewer than 700 patients were included, and this low number may have affected the statistical power and validity of the study. Second, this was a retrospective study using NHIRD data; therefore, the prescription adherence of each patient could not be assessed. That is, while the prescriptions of the patients could be identified through the health insurance database, the true drug dosage consumed by each patient could not be assessed. Third, it could not be fully guaranteed that the included patients used a nasogastric tube at each time point during the whole seven-month period because this retrospective study could only select patients based on the procedure code recorded in the NHIRD. Fourth, the indication bias discussed above may also have played an important role in the analysis of the results.

## Conclusion

According to the results of this study, there was no significant negative association between prokinetic agent use and pneumonia admission among the long-term nasogastric tube-dependent patients. There is no evidence supporting the prescription of prokinetic agents for pneumonia prevention in long-term nasogastric tube-dependent patients. Further prospective randomized controlled trials may be needed to examine the effects of prokinetic agents on pneumonia prevention among the long-term nasogastric tube-dependent patients.

## Supporting information

**S1 Table. The association between prokinetics exposure and pneumonia admission (time period changed to 14 days).**
(DOCX)

**S2 Table. The association between prokinetics exposure and pneumonia admission (time period changed to 45 days).**
(DOCX)

**S3 Table. The association between prokinetics exposure and pneumonia admission (washout period changed to 7 days).**
(DOCX)

**S4 Table. The association between prokinetics exposure and pneumonia admission (washout period changed to 15 days).**
(DOCX)

## Acknowledgments

The authors appreciate the Biostatistics Center, Kaohsiung Chang Gung Memorial Hospital, for their statistical work. The authors also wish to thank the Health Information and Epidemiology Laboratory at Chiayi Chang Gung Memorial Hospital for comments and assistance in performing the data analysis.

## Author Contributions

**Conceptualization:** Kun-Siang Huang, Wei-An Lai, Yao-Hsu Yang, Chia-Pei Chou.

**Data curation:** Kun-Siang Huang, Pin-Jie Bin.

**Formal analysis:** Pin-Jie Bin.

**Investigation:** Kun-Siang Huang.

**Methodology:** Kun-Siang Huang, Bo-Lin Pan, Wei-An Lai, Pin-Jie Bin, Yao-Hsu Yang.

**Project administration:** Kun-Siang Huang, Bo-Lin Pan, Chia-Pei Chou.

**Resources:** Chia-Pei Chou.

**Software:** Pin-Jie Bin, Yao-Hsu Yang.

**Supervision:** Chia-Pei Chou.

**Validation:** Pin-Jie Bin, Yao-Hsu Yang, Chia-Pei Chou.

**Visualization:** Wei-An Lai.

**Writing – original draft:** Kun-Siang Huang.

**Writing – review & editing:** Kun-Siang Huang, Chia-Pei Chou.

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
