## [Decision Letter · Decision Letter 0]

27 Jan 2021

PONE-D-20-36433

Could prokinetic agents protect long-term nasogastric tube-dependent patients from being hospitalized for pneumonia? A nationwide population-based case-crossover study

PLOS ONE

Dear Dr. Chou,

Thank you for submitting your manuscript to PLOS ONE. After careful consideration, we feel that it has merit but does not fully meet PLOS ONE’s publication criteria as it currently stands. Therefore, we invite you to submit a revised version of the manuscript that addresses the points raised during the review process.

We look forward to receiving your revised manuscript.

Kind regards,

Alexandru Rogobete, MD, PhD, MSc, ClinRes

Academic Editor

PLOS ONE

Journal Requirements:

Additional Editor Comments:

Dear Authors,

Please consider all the suggestions of the reviewers.

Please answer punctually for each suggestion and explain the change.

BR,

Reviewers' comments:

Reviewer's Responses to Questions

**Comments to the Author**

1. Is the manuscript technically sound, and do the data support the conclusions?

Reviewer #1: Yes

Reviewer #2: Yes

2. Has the statistical analysis been performed appropriately and rigorously? 

Reviewer #1: N/A

Reviewer #2: Yes

3. Have the authors made all data underlying the findings in their manuscript fully available?

Reviewer #1: Yes

Reviewer #2: Yes

4. Is the manuscript presented in an intelligible fashion and written in standard English?

Reviewer #1: Yes

Reviewer #2: Yes

5. Review Comments to the Author

Reviewer #1: The subject of the article is extremely important and actual.

I have the following comments and questions for the authors. There are many awkward phrases that I do not point out here; I only point out those where the meaning cannot be interpreted:

The table 1 is extremely hard to read! Try to make more readable.

The paragraph "Variable assessment" is not clear can be rewrite in more clear format.

The conclusion need to clear and specific.

My recommendation is to focus on 3 short conclusion!

Please recheck the References order.

Please double check the article by a native English reader.

Reviewer #2: In my opinion, this manuscript is clearly written and well structured. It does explain appropriately the rationale for the evaluation of the presented outcome: the need for hospitalisation for pneumonia considering, however, no other secondary outcomes, which I will refer further.

1. The authors defined 7- month period for nasogastric tube placement as being long-term period? Why was considered this time interval fur studying this issue?

2. A modern approach of the patients with neurodegenerative diseases, which implies deglutition problems is considered to be, nowadays, the percutaneous or laparoscopic gastrostomy with or without fundoplication. Considering that, the importance of the issue discussed here (the patients with long-term nasogastric tube-placement) is reduced even more. Additionally, it would be appropriate to analyse the PEG-patients (subsequently excluded for this study) vs. naso-gastric-tube dependent patients in terms of aspiration pneumonia.

3. Regarding lines 114-116, in my opinion, it would be valuable to reconsider the study design strategy in order to introduce as secondary outcome the number of pneumonia recurrences in this patients and not to eliminate completely this relevant clinical fact. Maybe prokinetics could reduce the incidence of the aspiration pneumonia episodes. A statistical analysis of this issue should be considered.

4. In accordance with table 1, mosapride was used in 56 patients only. Were there any statistical power tests conducted in order to evaluate the patients number needed for an accurate statistical analysis? If yes, it should be specified.

5. In accordance with FDA recommendation from 2017, metoclopramide should be used with caution in patients with Parkinson. What was the rationale to consider this subgroup of patients receiving this drug?

6. I would consider, as well, the timespan between the nasogastric tube insertion and the first aspiration pneumonia episode. It could be relevant as secondary outcome.

7. Another major concern is the inability of the study to identify the patient prescription adherence. It is indeed very difficult to consider this, but this is, in fact, the main purpose of this study: to determine if the prokinetic therapy influences the appearance of pneumonia episodes, who need hospitalisation.

6. PLOS authors have the option to publish the peer review history of their article (what does this mean?). If published, this will include your full peer review and any attached files.

Reviewer #1: No

Reviewer #2: No

---

## [Author Response · Author response to Decision Letter 0]

9 Mar 2021

Reply to reviewer #1

We appreciated a lot for your useful recommendation about our article. We apologized for poor English grammar and expression since English is not our native language. We have had our manuscript revised by native English reader. We would explain the revisions according to your precious suggestions as followings:

1. The table 1 is extremely hard to read! Try to make more readable.

Response:

Thank for your suggestion. We had modified the table 1 content to make it more readable. Please see the modified table 1 for details (line 211).

2. The paragraph "Variable assessment" is not clear can be rewrite in more clear format.

Response:

We appreciated your recommendation. We had revised our manuscript in variable assessment section to make it more fluent and readable. We also add confounding factors in the subtitle because in that paragraph, we also focused on explaining the confounding medications. Please refer to the Variable assessment and confounding factors section for more details (line 159-184).

3. The conclusion need to clear and specific. My recommendation is to focus on 3 short conclusion!

Response:

Thank you for your useful recommendation. We tried to make our conclusion objective and evidence-based according to results. Although the findings in our analyses revealed no significant results, there are still some implications for clinical practice. We had revised the conclusion section to make it clearer and more specific. Please refer to the conclusion section for details (line 333-340).

4. Please recheck the References order.

Response:

Thank for your carefully reminder. We have rechecked our reference order and found that in the original manuscript line 72, the reference 3 appeared after the reference 7 (in line 64). This is because reference 3 was also cited in line 64 which was expressed by “the presence of an endotracheal tube and enteral tube feeding[2-7].” We also revised the reference citation format to meet the PLOS one requirement. Please refer to the line 64 and 72 for further information and inform us if there were other mistakes that we don’t find. Thank you very much.

5. Please double check the article by a native English reader.

Response:

We appreciated your recommendation. We have invited a professional native English reader to recheck and revise English expressions in our manuscript for more clarity and fluency. The English revision certification would also attached in our uploaded files.

Reply to reviewer #2

We are so grateful for your precious comments and suggestions for our manuscript which were really clinically practical. We would reply to your suggestions as followings:

1. The authors defined 7- month period for nasogastric tube placement as being long-term period? Why was considered this time interval fur studying this issue?

Response:

Thank for your opinion and question. In this study, we used case-crossover design to do the within-object evaluation of prokinetic agent exposure and pneumonia admission in long-term nasogastric tube (NG tube) dependent patients. In fact, we presumed the long-term period as 3-month in our study design. The reason for selecting patients with NG tube placement for 7-month period before pneumonia admission was to confirm all the patients included were NG tube dependent at least 3 months among case period and control periods. Please refer to the Fig 2 in our manuscript (or below this paragraph). In Fig 2, the second control period (Control Period 2) was defined as the period between day -121 to day -150 and if we did not select patients had NG tube dependence 210 days before the index date, then the patient may not have NG tube dependence for 3 months in the second control period. For example, if a patient was NG tube dependent only 180 days before the index date, then at the second control period time point (the -121 day), the patient would only had used NG tube for 60 days before that control period date and which did not meet the long-term NG tube dependent presumption (3-month period) in our study. We had added the long-term NG tube dependent presumption in the Patient Selection section (line 117-125).

2. A modern approach of the patients with neurodegenerative diseases, which implies deglutition problems is considered to be, nowadays, the percutaneous or laparoscopic gastrostomy with or without fundoplication. Considering that, the importance of the issue discussed here (the patients with long-term nasogastric tube-placement) is reduced even more. Additionally, it would be appropriate to analyse the PEG-patients (subsequently excluded for this study) vs. naso-gastric-tube dependent patients in terms of aspiration pneumonia.

Response:

We are very appreciated for your opinion and useful suggestion. Indeed, the percutaneous or laparoscopic gastrostomy is getting more and more important in the neurodegenerative disease patients. However, in Taiwan, despite that PEG had been introduced in since 1995, the use rate of PEG was still low, and it may be explained by the traditional Chinese cultural value. Nasogastric tubes were used for a year or more in over half of the patients requiring enteral feeding in Taiwan(Yeh, Lo, Fetzer, & Chen, 2010). Under such social value in Taiwan, our article was aimed to evaluate the current clinical condition. Therefore, we still selected all the long-term NG tube dependent patients for analysis. What’s more, because the NG tube may be a risk factor of pneumonia occurrence by gastric organism migration along the tube to the pharynx(Gomes, Pisani, Macedo, & Campos, 2003). It may be more valuable and interesting to evaluation the prokinetic agent’s pharmacologic effect in pneumonia occurrence in these NG tube feeding patients rather than PEG dependent patients. However, it is still important in comparing the pneumonia risk between NG tube feeding patients and PEG dependent patients. Nevertheless, under our case crossover study design, the gastrostomy or PEG users were only 11 (in our manuscript Fig1, patient selection flow diagram). Hence, it was difficulty in analyzing the difference between NG tube users and PEG users in our study. It may be a very good and important implication in the future.

3. Regarding lines 114-116, in my opinion, it would be valuable to reconsider the study design strategy in order to introduce as secondary outcome the number of pneumonia recurrences in this patients and not to eliminate completely this relevant clinical fact. Maybe prokinetics could reduce the incidence of the aspiration pneumonia episodes. A statistical analysis of this issue should be considered.

Response:

We appreciated for your precious suggestion very much. Indeed, in cohort study design, it is important to consider the pneumonia occurrence episode in the data analysis between the exposure group and control group. However, in our manuscript, the case crossover study design was quite different from the cohort study design. The case crossover design was introduced by Maclure et. al(Maclure, 1991) and it is the case-only study design. This case crossover study design included all the participants with events (such as pneumonia admission in our study) and provided within subject comparison at different time points (case periods and control periods) for short-term effect of medicines or environmental exposure. Therefore, all included patients had pneumonia admission and we set the index date as the day of pneumonia admission. We included the first-time pneumonia admission patients in the database only to prevent from interfering of control periods and case period. To say, it is unable to analyze the pneumonia episodes as outcome evaluation under our case crossover study design in this nationwide database study because of the case crossover study designation nature and method. However, you still give us a very important and useful suggestion for future study design direction. We thank you very much. 

4. In accordance with table 1, mosapride was used in 56 patients only. Were there any statistical power tests conducted in order to evaluate the patients number needed for an accurate statistical analysis? If yes, it should be specified.

Response:

Thank you for your opinion and suggestion. Indeed, the mosapride users were only 56 patients and statistical power test should be conducted in cohort study to verify the statistical validity. However, in case crossover study, which is a retrospective case only study design comparing the same subject at different time point exposures, it is unusual to calculate the statistical power or the patient number needed to reach the statistical power. 

Regardless of that, we still conducted the statistical power test for the mosapride user group and the statistical power test for mosapride user group was only 43.50% which was far below 80%. We also conducted the statistical power test for all prokinetic agent, and the patient number needed to reach statistical power of 80% was 3,049. However, in our study, only 639 patients were included in this large scale nationwide longitudinal database consisted of 1 million people under the rigorous inclusion criteria for this case crossover design. To say, we have tried our best to extract all the patients we needed in this large database, but it was still difficult to reach the number needed for statistical power of 80%. It was almost impossible to increase included patient number in current large database and study design. We thought that It represented that we have interpret our results with caution, but the results of our study were still valuable for applying in the long-term NG tube dependent patients. Therefore, we have mentioned the small sample size in the manuscript “Limitation” section and revised the text to explain this problem (line 319-322). Again, we really appreciate your suggestion for the statistically problem.

5. In accordance with FDA recommendation from 2017, metoclopramide should be used with caution in patients with Parkinson. What was the rationale to consider this subgroup of patients receiving this drug?

Response:

Thank you for careful clinical suggestion and opinion. Certainly, according to the FDA recommendation, it was not suggested to use metoclopramide in patients with parkinsonism in case of extrapyramidal symptoms exacerbating the parkinsonism. However, in clinical practice, there may still some parkinsonism patients may have severe gastrointestinal tract symptoms like delayed emptying despite of other prokinetic agent use. Under such circumstances, metoclopramide may still be used with caution for symptom relief. What’s more, this study included patients’ data among 1997 and 2013 when the 2017 FDA recommendation not applied yet. Therefore, clinicians may still use metoclopramide in these parkinsonism patients after weighing the benefits and harms during that period of time.

6. I would consider, as well, the timespan between the nasogastric tube insertion and the first aspiration pneumonia episode. It could be relevant as secondary outcome.

Response:

We appreciated for your suggestion and opinion very much. It is really important and needed to consider the timespan between the NG tube insertion and the first pneumonia episode in cohort study design and it should provide some clinical useful information for data analysis or clinical implication. However, we have to apologize for inability to do such analysis. The main reason was somewhat like question 3, the study design method. We would like to restate that we used case crossover design as the basic foundation in our study. In this study design, we included all the patients with pneumonia admission and long-term NG tube dependence. We used conditional logistic regression to compare the drug exposures among case period and control periods WITHIN the same subjects at DIFFERENT time points. It is somewhat not that meaningful to compare the timespan between the nasogastric tube insertion and the first pneumonia episode because the case period and control periods are already set by study designer. For example, the NG tube insertion timespan would certainly be more at the case period (the time when pneumonia admission) than the first control period (day -61) by 60 days. The timespan between NG tube insertion and admission could not be analyzed in the case crossover design. Therefore, this is why the reason we set the 7-month NG tube dependence period as inclusion criteria to make sure that all the participants in control periods were NG tube dependence at least for 3-month period which was presumed to be the long-term NG tube dependence in this study. 

We hope our explanation would be clear enough, but we still really appreciated your professional suggestion to include the timespan as secondary outcome. We would take that in mind and may analyze the timespan in another future cohort study design. Thank you very much!

7. Another major concern is the inability of the study to identify the patient prescription adherence. It is indeed very difficult to consider this, but this is, in fact, the main purpose of this study: to determine if the prokinetic therapy influences the appearance of pneumonia episodes, who need hospitalisation.

Response: 

We appreciated for your comment on this important limitation of our study. In such kind of nationwide database study, the prescription adherence was somewhat an inherent limitation. Because we could only collect data or prescription status from the national health insurance database record, it is really difficulty to evaluate the realistic medicine consumption status. However, the advantage of large-scale database study is its large case number which may offset the bias of prescription adherence. A prior meta-analysis study included around 60 to 300 participants to explore the association of prokinetic agent and pneumonia in NG tube dependent patients(Liu, Dong, Yang, Wang, & Wang, 2017). Our study included more than 600 participants which was the most participants in this kind of studies evaluating the association of pneumonia and prokinetics in NG tube dependent patients. Therefore, we thought the results of our study was still valuable by its large case number despite the prescription adherence problem. We also mentioned the prescription adherence problem in our limitation section (line 322-326).

References:

Gomes, G. F., Pisani, J. C., Macedo, E. D., & Campos, A. C. (2003). The nasogastric feeding tube as a risk factor for aspiration and aspiration pneumonia. Curr Opin Clin Nutr Metab Care, 6(3), 327-333. doi:10.1097/01.mco.0000068970.34812.8b

Liu, Y., Dong, X., Yang, S., Wang, A., & Wang, M. (2017). Metoclopramide for preventing nosocomial pneumonia in patients fed via nasogastric tubes: a systematic review and meta-analysis of randomized controlled trials. Asia Pac J Clin Nutr, 26(5), 820-828. doi:10.6133/apjcn.102016.01

Maclure, M. (1991). The case-crossover design: a method for studying transient effects on the risk of acute events. Am J Epidemiol, 133(2), 144-153. doi:10.1093/oxfordjournals.aje.a115853

Yeh, L., Lo, L. H., Fetzer, S., & Chen, C. H. (2010). Limited PEG tube use: the experience of long-term care directions. J Clin Nurs, 19(19-20), 2897-2906. doi:10.1111/j.1365-2702.2009.03157.x

---

## [Editor Report · Decision Letter 1]

23 Mar 2021

Could prokinetic agents protect long-term nasogastric tube-dependent patients from being hospitalized for pneumonia? A nationwide population-based case-crossover study

PONE-D-20-36433R1

Dear Dr. Chou,

We’re pleased to inform you that your manuscript has been judged scientifically suitable for publication and will be formally accepted for publication once it meets all outstanding technical requirements.

Kind regards,

Alexandru Rogobete, MD, PhD, MSc, ClinRes

Academic Editor

PLOS ONE

---

## [Editor Report · Acceptance letter]

25 Mar 2021

PONE-D-20-36433R1 

Could prokinetic agents protect long-term nasogastric tube-dependent patients from being hospitalized for pneumonia? A nationwide population-based case-crossover study 

Dear Dr. Chou:

I'm pleased to inform you that your manuscript has been deemed suitable for publication in PLOS ONE. Congratulations! Your manuscript is now with our production department. 

Kind regards, 

on behalf of

Dr. Alexandru Rogobete 

Academic Editor

PLOS ONE